# Organizational Justice and Health: A Survey in Hospital Workers

**DOI:** 10.3390/ijerph19159739

**Published:** 2022-08-08

**Authors:** Nicola Magnavita, Carlo Chiorri, Daniela Acquadro Maran, Sergio Garbarino, Reparata Rosa Di Prinzio, Martina Gasbarri, Carmela Matera, Anna Cerrina, Maddalena Gabriele, Marcella Labella

**Affiliations:** 1Section of Occupational Medicine and Labor Law, Università Cattolica del Sacro Cuore, 00168 Roma, Italy; 2Department of Woman, Child & Public Health Sciences, Fondazione Policlinico Universitario A. Gemelli IRCCS, 00168 Roma, Italy; 3Department of Educational Sciences, University of Genova, 16126 Genova, Italy; 4WOW—Work and Organisational Well-Being Research Group, Department of Psychology, Università di Torino, 10124 Torino, Italy; 5Department of Neuroscience, Rehabilitation, Ophthalmology, Genetics and Maternal/Child Sciences (DINOGMI), University of Genoa, 16132 Genoa, Italy; 6Local Sanitary Unit Roma4, 00053 Civitavecchia, Italy

**Keywords:** mental health, occupational stress, job strain, back pain, healthcare, sickness absence, mediator analysis, work organization, occupational safety, occupational health

## Abstract

In complex systems such as hospitals, work organization can influence the level of occupational stress and, consequently, the physical and mental health of workers. Hospital healthcare workers were asked to complete a questionnaire during their regular occupational health examination, in order to assess the perceived level of organizational justice, and to verify whether it was associated with occupational stress, mental health, and absenteeism. The questionnaire included the Colquitt Organizational Justice (OJ) Scale, the Karasek/Theorell demand-control-support (DCS) questionnaire for occupational stress, and the General Health Questionnaire (GHQ12) for mental health. Workers were also required to indicate whether they had been absent because of back pain in the past year. Organizational justice was a significant predictor of occupational stress. Stress was a mediator in the relationship between justice and mental health. Occupational stress was more closely related to perceptions of lack of distributive justice than to perceptions of procedural, informational, and interpersonal justice. Physicians perceived significantly less distributive justice than other workers. In adjusted univariate logistic regression models, the perceptions of organizational justice were associated with a significant reduction in the risk of sick leave for back pain (OR 0.96; CI95% 0.94–0.99; *p* < 0.001), whereas occupational stress was associated with an increased risk of sick leave (OR 6.73; CI95% 2.02–22.40; *p* < 0.002). Work organization is a strong predictor of occupational stress and of mental and physical health among hospital employees.

## 1. Introduction

Organizational justice is a construct that defines the quality of social interaction in the workplace [1]. Organizational justice can be divided into four categories: procedural justice (fairness of decision-making procedures), distributive justice (fairness of outcomes), informational justice (correctness and completeness of information received in the workplace), and relational justice (equality and fairness in the interpersonal treatment of employees by their supervisors). Organizational justice is related to employee health and well-being. Low perceived equity has been shown to be associated with experienced stress reactions and related physiological and behavioral responses, such as inflammation, sleep problems, cardiovascular regulation, and cognitive impairment, as well as high rates of absenteeism from work [2]. Globally, low back pain was the leading cause of the number of years lived with disability in 2016 [3], and absenteeism due to this non-communicable disease is significantly associated with work-related physical demands [4] and psychosocial stress [5,6]. For this reason, perceptions of organizational justice may be associated with nonspecific back pain.

In complex systems, especially in work domains that are essential to our society, organizational structure may lack coherence and transparency, leading to perceived injustice [7]. The hospital is one of these complex structures where some workers may perceive a lack of organizational justice. The first type of complexity derives from the clinical variability of the patients. For some time now, scales have been developed that make it possible to measure the care complexity of patients and the intensity of the care they need. For example, the National Early Warning Score (NEWS) and the National Early Warning Score 2 (NEWS2) may identify patients at risk of in-hospital mortality and other adverse outcomes [8]. The intensity of care and assumed risk in treating medically complex patients should be taken into consideration in deciding health policy, reimbursement, and hospital resource allocation [9]. This patient-centered organizational approach overlaps the traditional hospital organization, characterized by numerous vertical operating structures and the specialties in which different professional figures (doctors, nurses, technicians, clerks, etc.) operate with different types of employment relationships (permanent employees, fixed-term contracts, freelancers, trainees, etc.). Furthermore, within each specialty, knowledge utilization and knowledge translation follow paths that depend on the context and process dynamics [10]. Managing these complexities is not easy. Healthcare professionals and hospital managers are often advised to learn from industry and businesses to improve quality and efficiency. However, there is often a lack of evidence that the implementation of industrial techniques and business methods improve the quality of care. Healthcare has a moral nature that cannot easily be organized into technological or corporate categories. For this reason, hospitals have been suggested to develop innovative management models, in which organizational justice contributes to defining the identity and intrinsic values of healthcare [11].

The perception of organizational injustice could be associated with increased occupational stress and physical and psychological symptoms in workers. In fact, organizational justice is one of the models that expresses the relationship between stressful factors present in the workplace and illnesses in workers [12,13,14,15]. Many workplace programs have been aimed at controlling psychosocial risks and promoting workers’ health and safety [16,17,18,19].

We aimed to test this hypothesis by asking healthcare workers (HCWs) in a public hospital to report their perceptions of organizational justice during the regular medical examination they undergo in application of the European guidelines for health and safety at work. Our aim was to assess the perceived level of organizational justice and to verify whether it was the same in the different categories of workers. We also wanted to investigate the relationship between organizational justice and occupational stress, workers’ mental health, and absenteeism due to low back pain.

## 2. Materials and Methods

In the first half of 2020, all workers in a hospital in Lazio, Italy, were sequentially asked to complete an anonymous questionnaire on organizational justice, occupational stress, and mental health, as part of the regular medical surveillance examination to which workers exposed to occupational risks are subjected.

The questionnaire included measures of organizational justice, stress, and mental health. Colquitt’s Organizational Justice Measure (OJM) [20], the Italian version [21], was developed to assess four aspects of justice: distributive justice, procedural justice, interpersonal justice, and informational justice. The OJM includes 20 items to be rated on a 5-point Likert scale with anchors of 1 = to a low degree and 5 = to a high degree. Typical questions are: “Have you been able to express your views and feelings concerning working procedures?” (procedural justice, 7 items); “To what extent does the benefits you get from work reflect the effort you have put into your work?” (distributive justice, 4 items); “To what extent did your supervisor or contact treat you in a polite manner?” (interpersonal justice, 4 items); “To what extent has your supervisor or contact person seemed to tailor (his or her) communications to individuals’ specific needs of the individual?” (informational justice, 5 questions). This instrument has been used in numerous studies, and psychometric studies have confirmed its 4-factor structure [22,23]. The Cronbach’s alphas in this study were 0.880 for distributive justice, 0.909 for procedural justice, 0.947 for interpersonal justice, and 0.883 for informational justice. The reliability of the overall score was 0.947.

The demand/control/support questionnaire (DCS) is a derivative of Karasek’s work content questionnaire [24], modified by Theorell [25], the Italian version [26]. The original questionnaire and the DCS scales used in different countries have similar characteristics [27]. The DCS questionnaire includes 17 questions to be rated on a 4-point Likert scale. Five items assess demand (i.e., the psychosocial stress of work; range 5–20); six items assess control (i.e., the components of occupational discretion and work coping skills; range 6–24); six items measure the support scale (i.e., perceived social support at work; range 6–24). Job strain can be calculated as a weighted relationship between demand and control. In this study, the reliability of the questionnaire subscales was 0.777 (demand), 0.615 (control), and 0.808 (support), respectively. A demand/control ratio of more than one is traditionally considered an indicator of excessive job stress. Excessive stress combined with social isolation (low support) is considered an indicator of isostrain, a condition with a high risk for mental and physical disorders.

The General Health Questionnaire (GHQ12) [28] is a measure of current mental health. In the Italian version [29], it includes 12 questions on topics related to anxious/depressive states and social dysfunction. Respondents are asked to indicate on a 4-point Likert scale (from 0 = “never” to 3 = “more than usual”) how often they have experienced each problem. In this way, a score from 0 to 36 is obtained, expressing the degree of mental disturbance. In this survey, the reliability of the questionnaire was 0.893 (Cronbach’s alpha).

Workers were also asked whether they had been absent from work in the past year because of back pain. The study is in accordance with the principles of the Declaration of Helsinki [30]. The study protocol was approved by the Ethics Committee of the Università Cattolica del Sacro Cuore, Rome, Italy (prot. N. 1226, 24 November 2016).

### Statistical Analysis

The distribution of the scores obtained from the questionnaires was initially studied using mean, median, and standard deviation We then compared the levels of perceived organizational justice and its components in the different categories of workers, in the two sexes, and in the different age groups. Comparisons were tested by one-way analysis of variance (One-way ANOVA) and, in case of significance, with multiple post-hoc comparisons using the Bonferroni correction for multiple comparisons.

To evaluate the relationship between justice, occupational stress, and mental health, we first studied the correlation between these continuous variables using the Pearson correlation coefficient. Once we verified the existence of a significant correlation between these three variables, we evaluated their relationship by hypothesizing that the perception of justice may act as a predictor, stress may be the mediator, and mental health (as measured by the GHQ12) as the response variable. In the mediation analysis, the level of confidence for all confidence intervals in output was set at 95%; the number of bootstrap samples for percentile bootstrap confidence intervals at 5000. We adjusted the mediation analysis by considering sex and age as covariates.

Logistic regression analysis was used to evaluate the association between justice or strain and having made absences for back pain in the year preceding the visit. The estimated effect was presented in terms of odds ratio (OR) and 95% confidence intervals. Each of the variables was initially posited as an independent variable in univariate models, in which the absence for back pain was the dependent variable.

The statistical analyses were carried out with the SPSS 26.0 program (IBM, Armonk, NY, USA) integrated by PROCESS vers. 4.1 [31].

## 3. Results

### 3.1. Characteristics of the Sample

Two-hundred and eighteen of two-hundred and forty-five health care workers submitted a complete questionnaire and were, thus, eligible for the survey. The participation rate was 89%. The refusal to participate was mainly motivated by the lack of time to complete the questionnaire. Almost two thirds of the workers were female. Most of them were over 40 years of age. Nurses were the largest group (Table 1). These characteristics correspond to those of the employees of the Italian National Health Service [32].

### 3.2. Intergroup Comparison

The distribution of perceptions of organizational justice among the different professional categories (physicians; nurses; and support staff, including technicians, clerks, and other staff), among genders, and among age groups is shown in Table 2. No significant differences in overall organizational justice ratings were found. Physicians, however, had significantly lower perceptions of distributive justice than auxiliary staff did. Older workers complained of significantly lower levels of informational justice than younger workers. Women reported lower levels of interpersonal justice than men.

### 3.3. Mediation Analysis

Organizational justice, occupational stress, and mental health were significantly correlated (Table 3). Lower levels of justice corresponded to higher levels of stress and mental disturbance, whereas stress and mental disturbance were positively related.

Mediation analysis revealed that both the direct and indirect (via job strain) associations between perceived justice with psychological disturbance were statistically significant, after adjustment for gender and age (Table 4). Figure 1 shows the complete set of standardized coefficients.

Logistic regression was used to examine the extent to which the perceptions of organizational justice were related to absenteeism due to back pain. In univariate models considering each variable separately, both organizational justice and social support were significantly associated with a lower risk of back pain, whereas job strain was associated with a significant increase in risk (Table 5). Next, age, sex, and job type were included in the model as control variables because their association with the incidence of back pain is well known. The effects of organizational justice, workload, and social support at work remained significant even after controlling for these variables.

## 4. Discussion

Our study has shown how, in a complex structure such as a hospital, organizational justice can be perceived very differently by workers. The findings indicated that the perceptions of organizational justice are closely related to occupational stress, and these two variables are directly and independently associated with psychological disorders. Organizational justice is also indirectly related to mental health through the mediation of job strain. Both justice and social support are important protective factors against back pain, whereas job strain is associated with a significantly increased risk of back pain.

Our findings are consistent with previous studies. A systematic review of longitudinal studies showed that procedural justice and relational justice were associated with mental health; these associations remained significant after controlling for occupational stress measured by both the DCS and other stress models [33]. These well-established observations have been confirmed by recent studies showing that among ICU nurses, there is an inversely significant association between perceived organizational justice and moral stress [34], and among hospital employees, organizational justice is a significant predictor of occupational stress [35].

As we have shown, the perceptions of equity vary across different groups in the hospital population. Physicians tend to complain about less distributive justice than other workers. Older workers perceive less informational justice. The latter aspect may be related to the fact that older people have greater difficulty using computer resources and digital media. Women who complain of less interpersonal justice than men indicate a gender gap. Gender gaps are not uncommon in the health professions, and often manifest in an unequal distribution of resources, information, and material rewards rather than explicit relational inequity [36], as reported in this case. In Italy, the Ministry of Labor and Social Policy has a National Equality Committee for the implementation of the principles of equal treatment and equal opportunities for workers, and many companies, including the one where we conducted our research, have local committees with the same objective.

Our investigation, therefore, confirmed the existence of some problems in the hospital organization that have been known and discussed for a long time, and for which various, but apparently ineffective, measures have been proposed. Greater transparency and fairness in the allocation of medical appointments, increasing electronic literacy and bridging the digital divide, and enforcing gender equity appear to be pervasive and achievable goals that can help improve the perceptions of organizational justice among health care workers.

Healthcare managers should have a vested interest in enhancing employees’ perceptions of equity as it relates to both health and productivity.

To our knowledge, this is the first study to show an association between organizational justice and absenteeism due to back pain. Many other negative effects of poor organizational justice have been reported in the recent literature. A prospective cohort study with an observation period of one year found that low interpersonal justice was associated with both the occurrence and duration of insomnia [37]. A Japanese study showed that female managers and professionals who suffer from low interpersonal justice in a workplace with unsupportive supervisors are more likely to engage in unhealthy coping behaviors to manage their stress [38]. Low perceived organizational justice is often associated with deviant behaviors [39,40]. Healthcare workers’ perceptions of organizational justice significantly predict turnover intention [41,42] and job satisfaction [43]. The relationship between distributive justice and turnover intention has been mediated by organizational commitment and work engagement [44]. The organizational justice perceived by healthcare workers significantly and positively affects their organizational trust and organizational identification [45]. Both distributive justice of appreciation and procedural justice contributed to lower productivity loss and sick leave after one year [46]. A study conducted during the COVID-19 pandemic suggests that reducing workload and promoting procedural justice are key factors that could be considered in interventions to reduce burnout risk among healthcare workers [47]. Reflection on the importance of organizational justice has prompted the incorporation of measures of relational justice and human factors into the algorithms used to assess the effects of hospital reorganization [48].

Work-related musculoskeletal disorders are one of the most common causes of occupational sick leave, accounting for 21% to 28% of work absenteeism days in 2017/2018 in some European countries [49]. In 2019, in the United States, back pain was one of the most prevalent conditions with aggregated medical costs [50]. Of the numerous ergonomic devices that are supposed to prevent back pain, some, such as insoles, have proven to be ineffective [51]. Meta-analyses also gave conflicting results: a study demonstrated that exercise combined with education and exercise alone both prevented back pain episodes and related absenteeism [52], whereas another systematic review concluded that physical exercise at the workplace did not reduce the occurrence of low back pain [53]. A consensus on the best methods for preventing back pain and absenteeism in healthcare workers must be reached. Our study adds to this debate a contribution that shows the importance of psychosocial factors.

This intriguing topic, therefore, requires well-conducted longitudinal studies, which also consider psychosocial factors. Our study deserves to be continued, observing in a prospective way the onset of back pain episodes and the relative absenteeism trend in relation to the different levels of perceived justice and occupational stress. Furthermore, monitoring the level of organizational justice perceived by workers can allow the occupational doctor to study and prevent phenomena such as job dissatisfaction, burnout, and turnover. What is more, it makes it possible to orient occupational health and safety services in a salutogenic sense, enhancing work engagement, job satisfaction, and the physical and mental well-being of workers.

Organizational justice may have a direct relationship with the quality of care. The lack of correctness in the organizational procedures (procedural justice) or in the transmission of the information necessary for the work (informational justice) could interfere with the diagnostic and therapeutic procedures that take place in the hospital. A future study could be designed to evaluate this hypothesis. The lack of interpersonal justice and distributive justice could also affect the motivations of health professionals, and, thus, indirectly harm patients. A longitudinal study could verify whether the justice perceived by hospital workers is associated with variations in the quality of care.

The results of this study should be considered in the light of some limitations. It was conducted with a cross-sectional design, which prevents causal inferences. A causal relationship between justice, occupational stress, and mental health could only be demonstrated by longitudinal studies. We hope that the continued health surveillance of these workers will allow such studies to be conducted in the future. Another limitation concerns the fact that the study was conducted in only one hospital, which limits the generalizability of the results. However, comparison with the literature suggests that the conditions in the public hospital where we conducted the study are similar to those in other hospitals.

## 5. Conclusions

In conclusion, our study has shown that hospital organization creates a sense of diminished distributive, interpersonal, and informational justice among a portion of the workforce. These organizational justice biases are associated with increased stress levels and poorer mental health, both directly and indirectly through job strain. In addition, perceived low justice is associated with absenteeism due to back pain. Hospital organization should be improved to increase workers’ health and the quality of care that depends on it.

The great complexity of the hospital structure and the conflicting organizational needs must not weigh on the health of workers, because it is obvious that, to guarantee the quality of care, the health and safety of health workers must first be guaranteed [54]. The complex management problems of hospitals cannot be addressed with solutions that harm workers or that they deem unfair. The pursuit of organizational justice should be one of the goals of good management.

Our study reinforces the view that the organization of the hospital must be based on criteria of justice, rather than on mechanisms aimed at maximizing production. To achieve this, a culture of respectful communication, justice in policies, and a proper procedure for allocating resources, workload, and rewards systems is a must.

## Figures and Tables

**Figure 1 ijerph-19-09739-f001:**
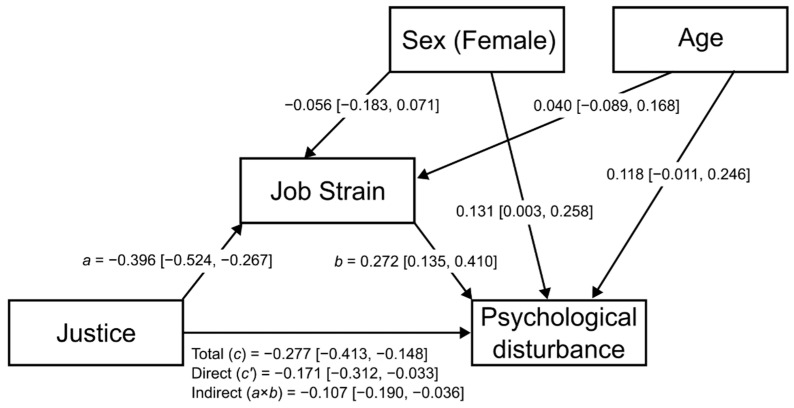
The mediation model for the direct and indirect (via job strain) associations of perceived justice with psychological disturbance, adjusting for age and sex (standardized coefficients and their 95% confidence interval).

**Table 1 ijerph-19-09739-t001:** Characteristics of the sample.

Sex	N	%
Male	83	38.1
Female	135	61.9
**Age**		
<25	9	4.1
30–34	13	6.0
35–39	43	19.7
40–44	36	16.5
45–49	41	18.8
50–59	68	31.2
>60	8	3.7
**Category**		
Physician	50	22.9
Nurse	112	51.4
Support staff	56	25.7

**Table 2 ijerph-19-09739-t002:** Distribution of perceived justice and its components (mean, standard deviation) by occupational group, gender, and age group.

	Physician (N = 50)	Nurse (N = 107)	Auxiliary Staff (N = 55)	*p* Value *
Organizational Justice	61.64 ± 16.61	63.09 ± 17.75	66.91 ± 15.98	0.206
Procedural Justice	19.44 ± 6.28	20.08 ± 6.39	20.87 ± 6.26	0.509
Distributive Justice	11.12 ± 3.62	11.84 ± 3.94	13.05 ± 4.04	0.036 1 vs. 3 = 0.035
Interpersonal Justice	14.74 ± 4.82	14.30 ± 3.95	15.47 ± 4.10	0.245
Informational Justice	16.34 ± 5.63	16.78 ± 4.38	17.51 ± 4.53	0.436
	**Younger (<40 years) (N = 65)**	**Middle (40–49 years) (N = 77)**	**Older (>50 years)** **(N = 76)**	
Organizational Justice	65.87 ± 14.41	65.55 ± 16.91	60.11 ± 16.02	0.053
Procedural Justice	20.33 ± 5.82	21.27 ± 6.74	18.80 ± 6.12	0.052
Distributive Justice	12.40 ± 3.60	12.38 ± 4.15	11.23 ± 3.94	0.124
Interpersonal Justice	15.35 ± 3.91	14.71 ± 4.13	14.16 ± 4.51	0.257
Informational Justice	17.76 ± 4.65	17.18 ± 4.49	15.80 ± 4.90	0.041 1 vs. 3 = 0.047
	**Male (N = 83)**	**Female (N = 129)**		
Organizational Justice	66.57 ± 14.45	61.92 ± 16.82		0.040
Procedural Justice	21.06 ± 6.09	19.55 ± 6.42		0.089
Distributive Justice	12.47 ± 3.53	11.68 ± 4.16		0.153
Interpersonal Justice	15.49 ± 3.58	14.21 ± 4.51		0.029
Informational Justice	17.54 ± 4.70	16.44 ± 4.72		0.096

Note: (*) One-way ANOVA with Bonferroni post-hoc test, or Student’s *t*-test.

**Table 3 ijerph-19-09739-t003:** Correlations between justice, job stress, and mental disturbance. Values in the lower triangle are zero-order correlations; values in the upper triangle are partial correlations after controlling for gender and age.

	1	2	3
1. Justice	1	−0.389 ***	−0.279 ***
2. Job Strain	−0.393 ***	1	0.342 ***
3. Mental disturbance	−0.304 ***	0.350 ***	1

Note: ***: *p* < 0.001.

**Table 4 ijerph-19-09739-t004:** Adjusted direct and indirect associations of justice with psychological disturbance (measured by GHQ12) mediated via job strain. Linear regression analysis and mediation analysis.

Measure	Unstandardized Coefficients	Standard Error	t	*p* Value	Standardized Coefficients
Total association	−0.094 (−0.138, −0.049)	0.022	−4.162	<0.001	−0.277 (−0.413, −0.148)
Direct association	−0.058 (−0.104, −0.011)	0.024	−2.441	0.015	−0.171 (−0.312, −0.033)
Indirect association mediated via Job strain	−0.036 (−0.069, −0.011)	0.015			−0.107 (−0.190, −0.036)
R^2^	0.183				

Note: Adjusted for age and sex.

**Table 5 ijerph-19-09739-t005:** Relationship between perceived justice, job strain, social support, and the risk of sickness absence for low back pain.

Variable	Model I (Unadjusted) OR (95% CI)	*p* Value	Model II (Adjusted) OR (95%CI)	*p* Value	*R^2^*
Organizational Justice	0.963 (0.943, 0.984)	<0.001	0.961 (0.939, 0.984)	<0.001	0.200
Job Strain	6.122 (2.009, 18.659)	<0.001	8.077 (2.385, 27.356)	<0.001	0.206
Social Support	0.843 (0.760, 0.936)	<0.001	0.816 (0.726, 0.918)	<0.001	0.209

Note: Model II was adjusted for age, sex, and job type.

## Data Availability

Data are available on Zenodo. DOI: https://doi.org/10.5281/zenodo.6810678 (accessed on 7 July 2022).

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
