# Peer review of "Organizational Justice and Health: A Survey in Hospital Workers"

_ijerph, 2022, doi:10.3390/ijerph19159739_

Round 1
Reviewer 1 Report
The authors aimed to assess the perceived level of organizational justice, and the relationship between organizational justice, occupational stress, and mental health in the group of medical workers. The authors emphasized the importance of back pain in employee absenteeism.
The purpose of the work is clearly defined. Previous research concerning the topic of the article has been sufficiently presented.
However, I would like to draw the authors' attention to several important issues:
- In the introduction part, I recommend to the authors add a paragraph explaining the concept of the complex organizational structure of a hospital. It is worth describing or listing at least some aspects of this complexity and the problems that result from it.
- Authors state: “Workers of a hospital in Lazio, Italy, were asked to complete a questionnaire (line 68)” and “Two hundred eight of 245 health care workers submitted a complete questionnaire 134 and were thus eligible for the survey” (lines 134-135). I would like to suggest the authors include more detailed characterization of the participants and present it in a table or text.
- It would be clearer to the reader if the authors would highlight particular subheadings in the results section, for example, intergroup differences or mediation analysis.
-I would like to point out that in the section Results the obtained results of statistical analyzes were not included in parentheses in the text of the article.
- I would recommend the authors compute and provide confidence intervals for each effect in the mediation analysis. The use of the confidence interval, especially for the indirect effect, indicates the acceptance or rejection of the null hypothesis.
- I suggest the authors present the mediation model graphically.
- In the discussion part authors state: “Perceptions of organizational justice are closely related to occupational stress, and these two variables are directly and independently associated with psychological disorders. Organizational justice is also indirectly related to mental health through the mediation of job strain. Both justice and social support are important protective factors against back pain, whereas job strain is associated with a significantly increased risk of back pain (lines 172-177)”. This is a duplication of the results. I recommend deleting or reformulating this paragraph.
- One of the main aspects of the paper was back pain and the associated absenteeism of workers. I would suggest the authors refer to this aspect to a greater extent in the discussion.
- I recommend to the authors widen the scope of conclusions.
Author Response
Reviewer #1
The authors aimed to assess the perceived level of organizational justice, and the relationship between organizational justice, occupational stress, and mental health in the group of medical workers. The authors emphasized the importance of back pain in employee absenteeism.
The purpose of the work is clearly defined. Previous research concerning the topic of the article has been sufficiently presented.
Response: We are very grateful to the reviewer for taking the time to review our work and thank him / her for appreciating it
However, I would like to draw the authors' attention to several important issues:
- In the introduction part, I recommend to the authors add a paragraph explaining the concept of the complex organizational structure of a hospital. It is worth describing or listing at least some aspects of this complexity and the problems that result from it.
Response: The hospital is a complex structure. The first type of complexity derives from the clinical variability of the patients. For some time now, scales have been developed that make it possible to measure the care complexity of patients and the intensity of the care they need. For example, the National Early Warning Score (NEWS) and the National Early Warning Score 2 (NEWS2) may identify patients at risk of in-hospital mortality and other adverse outcomes. The intensity of care and assumed risk in treating medically complex patients should be taken into consideration in deciding health policy, reimbursement, and hospital resource allocation. This patient-centered approach overlaps the traditional hospital organization, characterized by numerous vertical operating structures, the Specialties, in which different professional figures (doctors, nurses, technicians, clerks, etc.) operate with different types of employment relationships (permanent employees, fixed-term contracts, freelancers, trainees, etc.). Furthermore, within each Specialty, knowledge utilization and knowledge translation follow paths that depend on the context and process dynamics. Managing these complexities is not easy. Healthcare professionals and hospital managers are often advised to learn from industry and businesses to improve quality and efficiency. However, there is often a lack of evidence that the implementation of industrial techniques and business methods improve the quality of care. Healthcare has a moral nature that cannot easily be organized into technological or corporate categories. For this reason, hospitals have been invited to develop innovative management models, in which organizational justice contributes to defining the identity and intrinsic values of healthcare. We added these concepts into the manuscript
- Authors state: “Workers of a hospital in Lazio, Italy, were asked to complete a questionnaire (line 68)” and “Two hundred eight of 245 health care workers submitted a complete questionnaire 134 and were thus eligible for the survey” (lines 134-135). I would like to suggest the authors include more detailed characterization of the participants and present it in a table or text.
Response: Following the correct indication of the reviewer, we specified in the methods that the study took place in the first half of 2020 on all workers undergoing health surveillance. We then indicated the characteristics of the sample in the results, inserting a table with these data. Sorry for a typo, there were 218 workers.
- It would be clearer to the reader if the authors would highlight particular subheadings in the results section, for example, intergroup differences or mediation analysis.
Response: Following the suggestion of the reviewer, we used some sub-headings
-I would like to point out that in the section Results the obtained results of statistical analyzes were not included in parentheses in the text of the article.
Response: We can see the Reviewer's point, but we report the details of the results in the tables and in the figure, and we feel that reporting them in the text, too, would be redundant.
- I would recommend the authors compute and provide confidence intervals for each effect in the mediation analysis. The use of the confidence interval, especially for the indirect effect, indicates the acceptance or rejection of the null hypothesis.
Response: We added confidence intervals to table 3 along with standardized effects.
- I suggest the authors present the mediation model graphically.
Response: We added a Figure as requested.
- In the discussion part authors state: “Perceptions of organizational justice are closely related to occupational stress, and these two variables are directly and independently associated with psychological disorders. Organizational justice is also indirectly related to mental health through the mediation of job strain. Both justice and social support are important protective factors against back pain, whereas job strain is associated with a significantly increased risk of back pain (lines 172-177)”. This is a duplication of the results. I recommend deleting or reformulating this paragraph.
Response: We agree. According to Hoogenboom and Manske, “All results must first be described/presented and then discussed. Thus, the discussion should not simply be a repeat of the results section.” At the beginning of the discussion, it is necessary to summarize the results achieved in the study, placing them in a broader context and comparing them with the literature. We have reformulated the text, trying to emphasize this logical process, which is not a repetition.
Hoogenboom BJ, Manske RC. How to write a scientific article. Int J Sports Phys Ther. 2012 Oct;7(5):512-7.
- One of the main aspects of the paper was back pain and the associated absenteeism of workers. I would suggest the authors refer to this aspect to a greater extent in the discussion.
Response: We welcomed the reviewer's suggestion with great enthusiasm and added to the discussion a paragraph on back pain and absenteeism and another paragraph on possible developments in our research.
- I recommend to the authors widen the scope of conclusions.
Response: Accepting the invitation of both reviewers, we have rewritten the Conclusions.
Reviewer 2 Report
The Summary of the article is well done, however it should still be improved. It would be important for the authors to include in the abstract the Research Question, that is, what is the objective of this research.
I would recommend authors to include "Safety" in the Keywords. It would be great if this article appeared in the listings of researchers who are researching Occupational Safety, and this way this article would have a greater number of citations and visibility.
The Literature Review is very restricted, and the authors should improve. It would be important to help frame this authors' research (which is very interesting), in the occupational safety of workers. Showing that the concern with the organization of occupational safety services, is a global concern, and that there are other studies. As a suggestion, enter other articles, I can suggest to the authors for example the article "A study on the reality of Portuguese companies about work health and safety" (https://doi.org/10.1201/b16490).
In the Materials and Methods, the authors did not identify the research question. Why is this research important? What is the gap that exists in the scientific community that this research fills? It would be very important for the quality of the article if the authors were able to identify their Research Question.
What criteria did the authors set for the selection of the sample? Why these people? What is the relevance for this research? This information is very important to substantiate this research, so the authors should review and put this information in the article!
Statistically the article has a very interesting and consistent analysis of the results. A validation of the consistency of the data obtained was made through Cronbach's Alpha, which gave interesting values.
The conclusions are very poor and should be strongly improved by the authors. For example, it would be very important that the authors recommend future work. As I mentioned before, this research is very interesting, and it would be very good if the scientific community could give continuity to this research. It would be a great pity if this research would just stop here...
Author Response
Reviewer #2
The Summary of the article is well done, however it should still be improved. It would be important for the authors to include in the abstract the Research Question, that is, what is the objective of this research.
Response: The reviewer is right, we forgot to indicate the objectives of the study in the abstract. We corrected the abstract.
I would recommend authors to include "Safety" in the Keywords. It would be great if this article appeared in the listings of researchers who are researching Occupational Safety, and this way this article would have a greater number of citations and visibility.
Response: Great advice! We have done.
The Literature Review is very restricted, and the authors should improve. It would be important to help frame this authors' research (which is very interesting), in the occupational safety of workers. Showing that the concern with the organization of occupational safety services, is a global concern, and that there are other studies. As a suggestion, enter other articles, I can suggest to the authors for example the article "A study on the reality of Portuguese companies about work health and safety" (https://doi.org/10.1201/b16490).
Response: The subject we have dealt with has a very large literature. Accepting the invitation of the reviewer we have added some references, without, of course, being able to give an account of all the existing studies. We have tripled the references in the Introduction, also citing the chapter indicated by the reviewer.
Bastos, A.; Sá, J.C.; Fernandes, S.M. A study on the reality of Portuguese companies about work health and safety. In Occupational Safety and Hygiene, 2nd ed.; Arezes, P.; Baptista, J.S.; Barroso, M.P.; Carneiro, P. (eds); Taylor & Francis, London,UK, 2014; Volume II, pp. 687–691. Available at: https://www.researchgate.net/publication/300376480_A_study_on_the_reality_of_Portuguese_companies_about_work_health_and_safety
In the Materials and Methods, the authors did not identify the research question. Why is this research important? What is the gap that exists in the scientific community that this research fills? It would be very important for the quality of the article if the authors were able to identify their Research Question.
Response: We thank the reviewer for the suggestion. We indicated the purposes of the article in the final part of the Introduction, just before the Methods, and then resumed the discussion of these objectives in the final part of the manuscript. We also indicated the purposes of the study in the abstract.
What criteria did the authors set for the selection of the sample? Why these people? What is the relevance for this research? This information is very important to substantiate this research, so the authors should review and put this information in the article!
Response: The reviewer correctly pointed out missing information. We invited all workers who underwent a medical examination in the workplace in the first half of 2020 because they were exposed to an occupational risk to participate in the survey sequentially. We have added this information.
Statistically the article has a very interesting and consistent analysis of the results. A validation of the consistency of the data obtained was made through Cronbach's Alpha, which gave interesting values.
Response: We sincerely thank the reviewer for the appreciation and praise of our study.
The conclusions are very poor and should be strongly improved by the authors. For example, it would be very important that the authors recommend future work. As I mentioned before, this research is very interesting, and it would be very good if the scientific community could give continuity to this research. It would be a great pity if this research would just stop here...
Response: We very much welcomed the reviewer's invitation that is implicit in this observation to continue the studies in this field. We have included in the discussion a paragraph on the developments of this research. Accepting the invitation of both reviewers, we have rewritten the Conclusions.
Round 2
Reviewer 2 Report
The article has been greatly improved by the authors! Congratulations to the authors for the work done to improve the article!